# *N*-Glycosylation Patterns across the Age-Related Macular Degeneration Spectrum

**DOI:** 10.3390/molecules27061774

**Published:** 2022-03-08

**Authors:** Ivona Bućan, Jelena Škunca Herman, Iris Jerončić Tomić, Olga Gornik, Zoran Vatavuk, Kajo Bućan, Gordan Lauc, Ozren Polašek

**Affiliations:** 1Clinical Hospital Centre Split, 21000 Split, Croatia; ivona1993bucan@gmail.com (I.B.); kajobucan@gmail.com (K.B.); 2Clinical Hospital Centre Sisters of Mercy, 10000 Zagreb, Croatia; jskuncaherman@gmail.com (J.Š.H.); zvatavuk@hotmail.com (Z.V.); 3Department of Public Health, University of Split School of Medicine, 21000 Split, Croatia; iris.jeroncic@mefst.hr; 4Department of Ophthalmology, University of Split School of Medicine, 21000 Split, Croatia; olga.gornik@genos.hr; 5Genos Ltd., 10000 Zagreb, Croatia; glauc@genos.hr; 6Faculty of Pharmacy and Biochemistry, University of Zagreb, 10000 Zagreb, Croatia; 7Algebra LAB, Algebra University College, Ilica 242, 10000 Zagreb, Croatia

**Keywords:** age-related macular degeneration, *N*-glycans, prediction, biomarker, discovery

## Abstract

The pathogenesis of age-related macular degeneration (AMD) remains elusive, despite numerous research studies. Therefore, we aimed to investigate the changes of plasma and IgG-specific *N*-glycosylation across the disease severity spectrum. We examined 2835 subjects from the 10.001 Dalmatians project, originating from the isolated Croatian islands of Vis and Korčula. All subjects were classified into four groups, namely (i) bilateral AMD, (ii) unilateral AMD, (iii) early-onset drusen, and (iv) controls. We analysed plasma and IgG *N*-glycans measured by HPLC and their association with retinal fundus photographs. There were 106 (3.7%) detected cases of AMD; 66 of them were bilateral. In addition, 45 (0.9%) subjects were recorded as having early-onset retinal drusen. We detected several interesting differences across the analysed groups, suggesting that *N*-glycans can be used as a biomarker for AMD. Multivariate analysis suggested a significant decrease in the immunomodulatory bi-antennary glycan structures in unilateral AMD (adjusted odds ratio 0.43 (95% confidence interval 0.22–0.79)). We also detected a substantial increase in the pro-inflammatory tetra-antennary plasma glycans in bilateral AMD (7.90 (2.94–20.95)). Notably, some of these associations were not identified in the aggregated analysis, where all three disease stages were collapsed into a single category, suggesting the need for better-refined phenotypes and the use of disease severity stages in the analysis of more complex diseases. Age-related macular degeneration progression is characterised by the complex interplay of various mechanisms, some of which can be detected by measuring plasma and IgG *N*-glycans. As opposed to a simple case-control study, more advanced and refined study designs are needed to understand the pathogenesis of complex diseases.

## 1. Introduction

*N*-glycans are a class of molecules attached to various proteins. About 40% of all human proteins are glycosylated, meaning that glycans have an essential role in many biological and pathological mechanisms [1,2].

Previous studies have well established the role of glycans in ageing [3,4], suggesting that their composition may be used as the marker of biological vs. chronological ageing [5]. However, most of our understanding of their role as potential biomarkers for diseases originates from oncology [6,7,8], especially the field of colorectal cancer [9]. In most instances, they do not seem to have the sufficient ability for causal disease occurrence prediction, but in some cases, they can be a valuable biomarker of disease progression [10].

The central question related to the use of any biomarker from plasma, including *N*-glycans, is the ability of plasma-based measurements to reflect the pathogenetic processes in the remote target tissues. Under the assumption that the disease-related changes can be generalised, previous studies have identified increased branching of glycan structures and a greater abundance of tri- and tetra-antennary structures in association with autoimmune diseases, such as rheumatoid arthritis [11], pancreatitis and cholangitis [12,13], or chronic inflammation and diabetes [14]. However, generalised patternsof glycosylation may not provide a sufficient resolution to capture the relevant information on specific processes, requiring focus on the glycosylation of a specific protein. In this case, the most commonly used protein is the IgG [15]. The glycosylation patterns of IgG may provide even more precise insight into the function of the main class of molecules involved in immune regulation, providing a possibility for even more refined biomarker development opportunities [16,17,18,19].

The interest in exploring *N*-glycans’ role in AMD stems from the understanding that various proteoglycans have a role in retinal development and functioning [20,21,22], suggesting that their manipulation may even have beneficial therapeutic effects [23]. This was seen for NA3, a tri-antennary glycan without sialic acid, which was implied as having a protective role in AMD pathogenesis [24,25]. Furthermore, more complex structures and interactions between various classes of biological molecules such as lipids, proteins, and glycans might affect AMD development and prognosis [24,26,27]. However, none of the numerous molecules investigated thus far have yielded a beneficial biomarker for disease prediction and progress monitoring, preventing a more thorough understanding of the disease pathogenesis [28,29,30]; notably, previous studies often invoked hyperlipidaemia as a possible risk factor [31]. One interesting concept proposed regarding pathogenesis is para-inflammation, a process of adaptive tissue response, particularly retinal microglia and its complement system, to oxidative stress in which the purpose is to preserve tissue homeostasis and functionality, which is especially interesting in tissues whose functionality depends on non-proliferative cells and metabolically active cells, such as the macular retina [32,33]. The adaptive response to tissue stress can occur at three levels: tissue cells, local tissue immune system, and systemic immune system. If the level of tissue stress exceeds the reparatory capacity, then macrophages release cytokines and chemokines to recruit circulating monocytes, and when the tissue factors are released into circulation, they may activate the systemic immune system [34]. The net effect of all these changes has been implied in the pathogenesis of AMD, reflecting a complex and step-wise breakdown of regulatory mechanisms and resulting in high-grade inflammation [35,36].

Therefore, the aim of this study was to explore the possibility to use plasma and IgG *N*-glycan profiles are potential biomarkers for the development or progression of AMD. Instead of performing a simple case-control analysis, we opted to explore the pattern of *N*-glycan changes across various disease progression stages.

## 2. Results

There were 2835 subjects from the 10.001 Dalmatians project involved in this study. We detected 106 (3.7%) cases of AMD; 66 of them had bilateral AMD and 40 had the unilateral form of the disease. In addition, 45 (0.9%) subjects had early-onset drusen. The adjusted analysis of previously implied risk factors showed that pooled AMD cases were older and had an apparent lipid metabolism dysfunction (Table 1).

The next step of analysis suggested several glycan classes showing either group-wide or pair-wise differences across the disease stages compared to controls. Most notably, we detected some form of differences across the investigated groups for the following peaks; GP2, GP6-8, DG12, DG13, G0, IgG_GP3, IgG_GP4, IgG_GP6, IgG_GP14, IgG_GP15, IgG_GP18 and IgG_GP23 (Table 2; a complete set of results is available in the Appendix A).

The final analytic models were based on adjustments for age and sex, and the glycan peaks were significantly associated with the AMD in the previous analytic steps. The models had suggested a single significant result for a binary AMD (all groups merged vs. controls) and lack of any significant difference between the controls and the early-onset drusen (Table 3). In contrast, unilateral cases had significantly lower odds of the IgG GP18 peak, marginally lowered IgG GP4 and marginally elevated GP6 (Table 3). Lastly, the bilateral AMD had significantly higher odds for the DG13 peak, followed by increased levels of GP6 and G0 and marginally decreased GP2 (Table 3).

## 3. Discussion

These results show a complex nature of age-related macular degeneration progression. We detected activation of the anti-inflammatory mechanisms in early-onset drusen and dysregulation of immune regulation in unilateral and strong pro-inflammatory signals in bilateral disease. However, collapsing all these stages into a single category, AMD, yielded only one significant result. This finding clearly shows that the biomarker discovery studies should aim towards more refined phenotypes as possible to avoid missing the possible differences across the disease severity or progression spectrum.

The early-onset disease is denoted by the occurrence of drusen, which presents a collection of protein and lipids which are assumed not to be causal for AMD, but may serve as the early-stage disease marker [37]. The summarised pattern of changes in this study suggests a pattern on non-linear changes in the inflammatory signal, which was significantly lower than the controls in the bivariate analysis, but disappeared in the multivariate analysis, suggesting confounding effects. This suggests that early-stage drusen could trigger immune-suppressive and possibly even reparatory mechanisms that oppose inflammatory ones, which is probably substantially affected by the younger age of subjects who have it, which was described in previous studies [32,33]. The high glycol-conjugates composition of drusen [38] might explain some of the changes detected in this study, suggesting that glycans might be an interesting target for more advanced analysis of drusen pathogenesis. One possible way forward would be to analyse the larger cohort of the early-stage drusen, supplemented by multiple glycan measurements. This study design would provide an opportunity to explore the time-sensitive changes better and to establish the more refined role of glycan changes across the early stages of the disease progression.

The unilateral form of the disease was considered as the second progression step, where the criteria for AMD are met in only one eye, while the other one was spared. This might mean that the underlying risk for the AMD is lower than in bilateral disease or that this is only a transitory stage towards bilateral affection [39]. We detected a significant decline of the bi-antennary IgG GP18 peak, which was previously implied in the immunomodulatory effect [40,41,42,43,44]. In addition, there was strong correlation of the same glycan class with hyperlipidaemia [45], which was invoked as a possible risk factor for AMD in previous studies [31]. At the same time, we detected a significant increase in the GP6, which was previously associated with an increased risk for numerous diseases [44]. These results suggest that the unilateral disease stage may correspond to a situation of declining immune regulation, which no longer manages to balance out and to suppress pro-inflammatory mechanisms present in the more advanced disease stages. This idea was previously implied as an important ageing mechanism, suggesting that ageing may hamper the balance between pro- and anti-inflammatory mechanisms and cause numerous disadvantageous phenotypes [46]. However, in the case of AMD, it should be noted that ageing is not a direct cause of the disease but that additional pathogenetic mechanisms are required for a final phenotype to be expressed [47].

Finally, we detected the strongest change in the bilateral disease form, which was considered as the terminal disease stage. This stage is characterised by widespread inflammation, which leads to vision loss that results from numerous pathways activation, including the complement [48,49]. The pattern of glycans in this stage reflected an increase in the tetra-antennary structures (DG13), which were previously seen in numerous terminal-stage diseases, including cancer and low- and high-grade inflammation [50,51,52]. In addition, we detected an increase in agalactosylated glycans (G0), which was previously reported in rheumatoid arthritis [53] and systemic sclerosis [54].

The generalised pattern of changes in AMD to a degree resembled that of liver fibrosis [55], suggesting a certain level of similarity in different organs. In both instances, there are numerous stages of tissue failure, which seem to undergo initial stages of increased reparation, followed by the gradual loss of compensation and dysregulation of tissue and immune mechanisms, leading to terminal-stage breakdown and expression of the final phenotype. Therefore, one of the leading contextual findings of this study would be the need for more refined phenotype measurement, where studies should focus on the disease progression stage, and not only the binary presence or absence of disease. This can allow for better understanding, comparisons and further insight into the complex disease patterns, causes and progression.

While these results can only be considered as exploratory, future studies should definitely look into the role of glycans in AMD pathogenesis in more detail. An interesting approach is the comparison of glycans from various accessible non-invasive sources, such as tears [56,57]. Further limitations of this study include its cross-sectional nature, requiring a proper clinical follow-up study that can be based on multiple biological samples and time points, as well as the use of other biomarkers for better tracking of disease status and progression.

Nevertheless, the results of this study demonstrate that plasma or IgG glycans can be used as biomarkers for AMD progression. Additionally, this study clearly shows the need for a more refined disease phenotype assessment and understanding in biomarker discovery studies, especially in the case of complex chronic diseases.

## 4. Materials and Methods

### 4.1. Setting

This study was based on the 10.001 Dalmatians project, which had the principal goal of exploring the genetic and environmental risk factors for health and disease in isolated human populations [58,59,60]. These included the inhabitants of two remote islands, the island of Vis (n = 591) and Korčula (n = 1427), complemented by the subjects recruited from the mainland, the coastal city of Split (n = 817). All subjects were initially informed about the study goals and procedures, after which they signed informed consent prior to inclusion. The study was approved by the ethics boards of the Medical School, University of Zagreb, the Multi-Centre Research Ethics Committee for Scotland, The University of Split School of Medicine and the Lothian NHS Board and was performed in the best research practices [61].

### 4.2. Subjects

For purposes of this study, we included the subjects who had bilateral retinal fundus photographs, alongside blood samples used for plasma and IgG *N*-glycan analysis. In addition, we used the available information on the most important risk factors, which were previously implied in AMD pathogenesis. These included socioeconomic estimates (years of schooling and a composite index for the material status estimation) and relevant lifestyle information, including smoking (classified as smokers and non-smokers; all individuals who reported having ceased smoking within the past five years were considered as active smokers) and self-reported level of physical activity. We also included several clinically relevant measurements in the analysis: serum measurements of the four core lipids (cholesterol, LDL, HLD and triglycerides), uric acid, glucose and HbA1c. In addition, medical history was used to extract information on hypertension (any subject with a measured blood pressure over 140 mmHg for systolic or 90 mmHg for diastolic, who had hypertension in their medical history, or who reported the use of antihypertensive medication), diabetes (in medical history), gout (in medical history), glaucoma (in medical history) or high myopia (in medical history). Lastly, we also used Sphygmocor to measure four advanced indicators related to cardiovascular status, namely the augmentation index, pulse-wave velocity, and systolic and diastolic central pressures.

### 4.3. Retinal Photography

A fixed digital fundus camera Canon iDRG was used in a sitting position to take a native photo of the fundus of both non-mydriatic eyes in a dark room. Subject preparation included sitting in a darkened room for an average of 10–20 min or as long as the eye needed for complete adaptation from extreme light to complete darkness to achieve maximum image quality of the ocular background. After adaptation to the dark, the subjects were fixed in a standard sitting position behind a camera, with chin and forehead resting on the base and looking at the object of fixation. This was followed by fundus photography with coloured filters where the retina was lit up by white light and explored in full colour.

### 4.4. AMD Scoring Scheme

The colour fundus photographs with a visible macular area were analysed according to the International Classification (IC) for AMD from 1995 [62]. According to this classification, the grading of AMD is based on colour fundus photography; visual acuity is not taken into consideration, and two forms of such grading are to be distinguished: (a) the early-stage disease with the presence of drusen (products of retinal metabolism and damaged cells between the RPE and the inner collagenous layer of the Bruch’s membrane) and/or abnormalities of RPE (hyper/hypopigmentation changes and depigmentation of RPE with faintly visible choroidal blood vessels) and (b) the late stage with the neovascular wet form with haemorrhages or dry form of geographic atrophy (depigmentation changes of RPE with clearly visible choroidal blood vessels). Participants who had other signs of retinal diseases, such as degenerative changes due to high myopia, chorioretinitis, diabetic retinopathy with laser photocoagulation as therapy, malignancy, macular rupture, or occluded arteries and veins, were excluded from research.

### 4.5. Glycan Measurements

Glycans were measured using a standardised protocol [2]. In brief, we used a combination of the HPLC analysis of fluorescently labelled glycans. After sialidase digestion, glycans were separated into 29 chromatographic peaks, which corresponded to mixtures of similar molecules in the peak, or in some cases, even corresponded to a single specific molecular component per peak. In addition, there were 17 derived traits, which were grouped to better reflect the glycan composition, meaning that the analyses included 46 plasma *N*-glycan fractions.

In the case of IgG *N*-glycans, we first isolated IgG from plasma by affinity chromatography [15]. Following this, IgG was denatured with SDS and then analysed similarly to plasma *N*-glycans [2]. The data for IgG glycans were based on 24 fractions, meaning that there were 68 *N*-glycan peaks used in the analysis.

### 4.6. Statistical Analysis

The numerical data were initially reported as the means and standard deviation, while the categorical data were reported as number and percent. The bivariate analysis was based on the analysis of variance (ANOVA), accompanied by pair-wise comparisons in post hoc testing (using Bonferroni’s post hoc test). The second analytic stage was based on the known risk factors of the AMD and utilised the logistic regression, where all AMD cases were considered as one group (regardless of their unilateral or bilateral presence) and the controls as the second. Finally, the last analytic stage stratified cases into three groups and compared them to controls. For that purpose, we used logistic regression and created one model for all three disease stages in the aggregated form (cases vs. controls) and a separate model for all three disease stages. The stages included early drusen as the earliest stage, unilateral as an intermediate, and bilateral as the terminal disease stage. In addition, we added age and sex as possible confounders in this model, along with the two lipid fractions (namely, HDL and LDL lipids), which were identified as the significant predictors in the second analytic step. All analyses were performed in R, with significance set at *p* < 0.05.

## Figures and Tables

**Table 1 molecules-27-01774-t001:** The adjusted odds ratios for the conventional risk factors for AMD, logistic regression, with odds ratios (OR) and 95% confidence intervals (CI).

Predictor	*p*	OR [95% CI]
Age (years)	<0.001	1.04 [1.02–1.06]
Sex		
Men (Ref.)		1.00
Women	0.065	1.47 [0.98–2.23]
Years of schooling (years)	0.101	1.05 [0.99–1.11]
Material status (composite index)	0.632	0.98 [0.92–1.05]
Body mass index	0.926	1.00 [0.96–1.05]
Smoking (yes/no)	0.883	1.03 [0.68–1.56]
Hypertension (measured or medical history)	0.795	1.05 [0.72–1.55]
Diabetes (medication or medical history)	0.403	0.71 [0.31–1.60]
Augmentation index	0.289	0.99 [0.96–1.01]
Pulse wave velocity	0.872	1.01 [0.92–1.11]
Central systolic blood pressure	0.880	1.00 [0.98–1.02]
Central diastolic blood pressure	0.309	1.01 [0.99–1.04]
Serum uric acid	0.496	1.01 [1.00–1.02]
Serum glucose	0.662	0.97 [0.82–1.13]
Gout (medication or medical history)	0.135	0.48 [0.18–1.26]
Glaucoma (medication or medical history)	0.143	1.79 [0.82–3.90]
Self-reported physical activity level	0.977	1.00 [0.79–1.27]
Serum HbA1C	0.113	0.81 [0.63–1.05]
Serum cholesterol	0.001	2.03 [1.35–3.05]
Serum triglycerides	<0.001	1.28 [1.11–1.46]
Serum HDL	0.005	2.41 [1.30–4.46]
Serum LDL	0.001	2.13 [1.39–3.27]

**Table 2 molecules-27-01774-t002:** The selection of *N*-glycans that had significant differences from the control group across the disease stages and controls (the entire set is provided in Appendix A).

Glycan Peak	Bilateral AMD (B)	Unilateral AMD (U)	Early-Onset Drusen (e)	Controls (c)	*p* (F)	Pair-Wise Significance **
GP2 *	4.13 ± 1.23 (1.90–7.14)	4.35 ± 1.25 (3.01–6.37)	2.75 ± 0.81 (1.59–4.66)	3.74 ± 1.39 (1.06–12.65)	0.009 (3.84)	-; -; 0.033
GP6 *	3.44 ± 0.74 (2.13–4.91)	3.78 ± 0.51 (3.27–4.85)	4.28 ± 1.00 (2.82–6.41)	3.77 ± 0.91 (1.49–7.39)	0.043 (2.72)	
GP7	11.97 ± 4.08 (8.32–26.34)	10.01 ± 1.1 (8.17–11.42)	10.00 ± 1.26 (8.14–13.15)	10.75 ± 2.42 (6.53–29.18)	0.038 (2.82)	
GP8 *	8.8 ± 1.19 (6.06–11.01)	9.42 ± 0.87 (7.86–10.39)	10.43 ± 2.10 (7.45–16.51)	9.41 ± 1.55 (5.74–14.93)	0.016 (3.45)	
DG12	0.93 ± 0.46 (0.42–2.29)	0.78 ± 0.20 (0.36–1.06)	0.73 ± 0.25 (0.39–1.14)	0.75 ± 0.30 (0.10–2.46)	0.029 (3.03)	0.017; -; -
DG13	0.95 ± 0.57 (0.32–2.30)	0.70 ± 0.19 (0.34–0.97)	0.60 ± 0.24 (0.28–1.15)	0.68 ± 0.28 (0.18–1.95)	<0.001 (7.75)	<0.001; -; -
G0	4.48 ± 1.20 (2.23–7.38)	4.67 ± 1.18 (3.41–6.76)	3.01 ± 0.81 (1.77–4.83)	4.07 ± 1.36 (1.39–12.84)	0.004 (4.42)	-; -; 0.017
IgG_GP3	0.12 ± 0.02 (0.08–0.15)	0.11 ± 0.05 (0.04–0.18)	0.08 ± 0.01 (0.07–0.09)	0.10 ± 0.03 (0.03–0.18)	0.049 (3.05)	
IgG_GP4	25.22 ± 5.54 (13.66–39.43)	22.52 ± 4.02 (18.57–29.68)	16.19 ± 4.88 (6.93–23.3)	20.99 ± 6.14 (6.48–47.89)	<0.001 (9.69)	<0.001; -; 0.004;
IgG_GP6	6.58 ± 1.60 (4.60–12.08)	6.37 ± 1.35 (4.78–8.93)	4.43 ± 1.10 (2.42–6.17)	5.61 ± 1.63 (2.10–12.86)	<0.001 (8.17)	0.003; 0.019; 0.010
IgG_GP14	9.75 ± 2.51 (4.81–17.54)	9.30 ± 2.29 (6.84–13.72)	14.59 ± 3.45 (8.74–20.94)	11.6 ± 3.67 (3.39–25.82)	<0.001 (8.50)	<0.001; 0.002; 0.002
IgG_GP15	1.41 ± 0.28 (0.76–2.31)	1.50 ± 0.23 (1.14–1.80)	1.76 ± 0.38 (1.14–2.49)	1.55 ± 0.36 (0.75–3.54)	0.009 (3.85)	
IgG_GP18	7.69 ± 1.63 (4.39–12.05)	7.20 ± 1.51 (5.05–9.86)	11.27 ± 2.65 (6.47–17.31)	9.07 ± 2.53 (3.29–19.38)	<0.001 (10.06)	0.008; -; 0.001
IgG_GP23	1.83 ± 0.57 (0.86–3.43)	1.79 ± 0.34 (1.18–2.23)	2.25 ± 0.50 (1.31–3.32)	2.07 ± 0.63 (0.70–4.67)	0.040 (2.79)	

* the glycan structure in these peaks correspond to peaks DG2, DG6 and DG8, which were removed from this Table since they were equal to the corresponding GP peaks; ** pair-wise comparisons across groups, where the first value denotes the comparison of bilateral AMD with controls, the second unilateral vs. controls, and the third one is the *post-hoc* test for the early onset vs. controls; significant values are shown.

**Table 3 molecules-27-01774-t003:** The results of multinomial logistic regression; only the significant results of the terminal step are shown.

Predictor	Controls vs. All Disease Stages; P; OR (95% CI)	Controls vs. Early-Onset; P; OR (95% CI)	Controls vs. Unilateral; P; OR (95% CI)	Controls vs. Bilateral; P; OR (95% CI)
GP2	-			0.049; 0.61 (0.39–1.16)
GP6	-	-	0.045; 2.42 (1.01–5.65)	-
GP7	-		-	0.012; 1.16 (1.02–1.30)
DG13	0.008; 2.96 (1.28–6.43)	-	-	<0.001; 7.90 (2.94–20.95)
G0	-	-	-	0.047; 1.83 (0.89–3.14)
IgG_GP4	-	-	0.043; 0.83 (0.68–0.99)	-
IgG_GP18	-	-	0.009; 0.43 (0.22–0.79)	-

## Data Availability

Data are available upon reasonable request from the corresponding author.

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
