# Peer review of "N-Glycosylation Patterns across the Age-Related Macular Degeneration Spectrum"

_molecules, 2022, doi:10.3390/molecules27061774_

Round 1

Reviewer 1 Report

The present study is aimed at analysing N-glycosilation of serum and IgG proteins in order to identify disease state or disease stage specific markers in a cohort of patients with AMD from the 10K Dalmatians project.

Comments:

Results

1) Line 85-87: percentages are in this paragraph are sometimes expressed in relation to the 2,835 subjects from the Dalmatians project and in others in relation to the total number of cases with AMD. The authors can either include a clarification next to each percentage or group the comparisons to the total number of subjects (AMD vs control) in one sentence then in the next sentence provide the breakdown of AMD cases to the  3 stages

2) Line 88,89: “The adjusted analysis of previously implied risk factors showed that  pooled AMD cases were older and had an apparent lipid metabolism dysfunction”

- Could the authors provide evidence or cite pervious studies that support the involvement of these risk factors in AMD?

- Could the authors provide justification for why they selected these risk factors to look at?

-  The age comparison referred to in the sentence is conducted between AMD and whom?

 3) Table1: repetition of three rows (gout, glaucoma and self-reported physical activity)

4) Table 2: “The selection of N-glycans that had significant difference from the control group across the four disease stages”

- There are only 3 stages of the disease

- How many cases in which each glycan peak was detected (n=?)

- What does (F) signify?

- The last two columns in the table are jumbled, it is difficult to discern which P value corresponds to the comparison between each disease  stage and the control and which corresponds to the pair-wise comparison done between disease stages only (i.e B vs U, B vs e and U Vs e) note that B vs U is missing here.

- Also which statistical test was applied for the control vs disease stages comparisons?

- Some of the glycan peaks with statistically significant alterations (GP3, DG7 and IgG_GP23, Supplemental Table1) are not presented in Table 2, is there a reason for excluding them?

- Table 2 footnote: the justification for omitting DG2,6,8 from the table is not clear? were they omitted because they share peak structural similarities with GP2,6,8? If this is the reason, it may be true for GP2/DG2 but not for DG6, DG 8 as they do not have overlapping features with any of the glycan peaks listed in Table 2 (see ref [2] Table 1)

4) Table 3: is mislabelled as Table 1

Discussion

1) Line 119-122:

- Could the authors specify which of the detected glycans are associated with anti-inflammatory or pro-inflammatory responses and which are implicated in the regulation of the direction of the immune response (cite references)

- The authors conclud that the detected glycan peaks are an indication of active immune responses (pro and anti-inflammatory), however, they do not provide data supporting the notion that  identification of glycan peaks would necessarily reflect the activation  of the pathway(s) they are involved in.

2) Line 129-131:

- The sentence is not clear. It is also missing reference to the the data being discussed.

-  Which type of inflammatory signal (pro or anti) that was observed to be reduced?

3) Line 131-134:

- Can the authors include a description of the information cited in the ref 32,33 that supports their claim?

- The use of the word mediate with young age is odd, did the authors mean to say that the anti-inflammatory responses that they predict to be acting in the early stage are active in young cases vs older ones?

5) Line 155: Can the authors indicate which glycans exactly were altered and in which pathways do they serve, or the function and biological role of the proteins they are conjugated to?

Methods

1) Line 184: Citation of a reference describing the 10K Dalmatians project is missing

2) The age range of the enrolled subjects and gender distribution is not provided.

3) Line 196-202: What is the basis upon which the authors determined that the motioned factors are the most important risk factors for AMD? Can the authors include supporting references?

4) Glycan isolation and measurements were done in what type of biospecimen, serum or plasma?

Author Response

We want to thank the reviewer for careful and dedicated reading of the manuscript, identification of problems and suggestions provided; we have accepted and addressed all of your comments below.

Results

1) Line 85-87: percentages are in this paragraph are sometimes expressed in relation to the 2,835 subjects from the Dalmatians project and in others in relation to the total number of cases with AMD. The authors can either include a clarification next to each percentage or group the comparisons to the total number of subjects (AMD vs control) in one sentence then in the next sentence provide the breakdown of AMD cases to the  3 stages

R: This was not very clear, and we would like to thank the reviewer for pointing this up. We removed the percentages that were not referring to the total sample size and made it clearer

2) Line 88,89: “The adjusted analysis of previously implied risk factors showed that  pooled AMD cases were older and had an apparent lipid metabolism dysfunction”. Could the authors provide evidence or cite previous studies that support the involvement of these risk factors in AMD? Could the authors provide justification for why they selected these risk factors to look at?

R: We split the previous references by the risk factors and add a sentence that suggests hyperlipidaemia as an important risk factor (reference 31)

-  The age comparison referred to in the sentence is conducted between AMD and whom?

R: this was already answered in comment 1; we changed the text accordingly, and it is now more readable.

3) Table1: repetition of three rows (gout, glaucoma and self-reported physical activity)

R: Thank you for detecting this; we deleted the three repetitive rows

4) Table 2: “The selection of N-glycans that had significant difference from the control group across the four disease stages”

- There are only 3 stages of the disease

R: This was an omission, we meant to say four groups, indeed, there were only three groups plus controls; corrected

- How many cases in which each glycan peak was detected (n=?)

R: All the cases were detected in all glycan peaks, it was just a matter of quantifying the peak height, which was used as the signal strength in this case

- What does (F) signify?

R: This is the test statistics for the ANOVA

- The last two columns in the table are jumbled, it is difficult to discern which P value corresponds to the comparison between each disease  stage and the control and which corresponds to the pair-wise comparison done between disease stages only (i.e B vs U, B vs e and U Vs e) note that B vs U is missing here.

R: This was the best way to show all the information, but indeed, it was jumbled. We have now removed all the excessive comparisons and left only three possible numbers to denote the comparison of controls with bilateral, unilateral and early-onset disease. The entire table is shown as Supplementary Table 1

- Also which statistical test was applied for the control vs disease stages comparisons?

R: Added to the Material and methods section

- Some of the glycan peaks with statistically significant alterations (GP3, DG7 and IgG_GP23, Supplemental Table1) are not presented in Table 2, is there a reason for excluding them?

R: These significances were marginal, and later not significant in the pairwise comparisons, so we excluded them. However, the review is right, if we had defined one approach, we could not have changed it along the way. The additional rows are now shown in Table 2, and then the entire set is also available n the Supplementary Table.

- Table 2 footnote: the justification for omitting DG2,6,8 from the table is not clear? were they omitted because they share peak structural similarities with GP2,6,8? If this is the reason, it may be true for GP2/DG2 but not for DG6, DG 8 as they do not have overlapping features with any of the glycan peaks listed in Table 2 (see ref [2] Table 1)

R: The numerical values of these peaks are the same; although there are some structures that do not exhibit a complete overlap, their abundance is non-detectable by the method, leading to the completely same numerical values; these remained excluded

4) Table 3: is mislabelled as Table 1

R: Corrected

Discussion

1) Line 119-122:

- Could the authors specify which of the detected glycans are associated with anti-inflammatory or pro-inflammatory responses and which are implicated in the regulation of the direction of the immune response (cite references)

R: We have now expanded the Discussion section by additional references and more clearly indicated which glycan structure has previously been linked with what outcomes. We want to thank the reviewer, as the Discussion is now broader and more suited for the paper.

- The authors conclud that the detected glycan peaks are an indication of active immune responses (pro and anti-inflammatory), however, they do not provide data supporting the notion that  identification of glycan peaks would necessarily reflect the activation  of the pathway(s) they are involved in.

R: This was based on epidemiological data and studies, which are now included in the Discussion section (references 44-50)

2) Line 129-131:

- The sentence is not clear. It is also missing reference to the the data being discussed.

R: corrected

-  Which type of inflammatory signal (pro or anti) that was observed to be reduced?

R: Corrected across the glycan peaks

3) Line 131-134:

- Can the authors include a description of the information cited in the ref 32,33 that supports their claim?

R: These two studies are explaining the para-inflammation, and this was already mentioned in the Introduction. We have now broadened this part.

- The use of the word mediate with young age is odd, did the authors mean to say that the anti-inflammatory responses that they predict to be acting in the early stage are active in young cases vs

older ones?

R: changed the word to affected

5) Line 155: Can the authors indicate which glycans exactly were altered and in which pathways do they serve, or the function and biological role of the proteins they are conjugated to?

R: This information is still not available at the molecular level, we only have the epidemiological data, listed in several references mentioned before. We have linked some of these to e.g. hyperlipidaemia, but for most of these peaks, we do not know the exact pathogenetic mechanisms

Methods

1) Line 184: Citation of a reference describing the 10K Dalmatians project is missing

R: we have added new references

2) The age range of the enrolled subjects and gender distribution is not provided

R: added in the Results text

3) Line 196-202: What is the basis upon which the authors determined that the motioned factors are the most important risk factors for AMD? Can the authors include supporting references?

R: yes, these are the risk factors identified in previous studies, including a study 31, in addition to the common confounders, age and sex.

4) Glycan isolation and measurements were done in what type of biospecimen, serum or plasma?

R: thank you for this, we made several omissions throughout the text – all the glycans were based on plasma; we have corrected this

Reviewer 2 Report

The manuscript by Bućan and colleagues seeks to investigate whether there are any serum N-glycosylation patterns that could be used as a predictive biomarker for age-related macular degeneration (AMD). They compare the N-glycosylation patterns of a single molecule, IgG, between serum from 106 AMD patients, 45 pre-AMD patients, and 2684 control subjects. Yet, several conclusions of the paper do not appear to be fully supported by their results, as the underlying statistical analysis reveals very little, if any, real significant differences amongst conditions. My concerns are as follows:

  • While there appear to be some statistically significant differences amongst conditions in their initial analysis (that did not account for age or sex), these results do not seem particularly convincing in a biological context. For example, for many of these changes, the direction of change in early-onset drusen versus diagnosed AMD patients is completely different. If these changes were real, shouldn’t they change in the same direction? Nonetheless, many of their conclusions come from this analysis, whereas more focus should be put on the controlled analysis.

  • In their second analysis, after accounting for age and sex, there are no significant differences between the controls and the early-onset drusen groups. Further, there are only two significant glycan changes in this analysis: IgG_GP18 for unilateral AMD and DG13 for bilateral AMD. Yet, the IgG_GP18 mark is no different in bilateral AMD, which casts doubt as to whether this difference is truly real.

  • The authors concede that “collapsing all three disease stages into a single outcome category (broad-sense AMD) did not yield a single significant result”. Yet, this appears to be buried, although this seems to be a key take-away. This should be bluntly stated, including in the abstract.

  • The authors make several conclusions to try to increase the significance of their results, which I do not find to be supported, such as:
    1. “The results suggested the anti-inflammatory changes in glycans in the early-stage drusen, breakdown of immune regulation in unilateral disease, and a substantial increase in the pro-inflammatory mechanisms in the bilateral disease occurrence”
    2. “The pattern of changes in this study revealed an interesting finding or lowered overall inflammatory signal”
    3. “Nevertheless, the results of this study demonstrate that serum or IgG glycans can be used as biomarkers for AMD progression”

  • The description of their statistical analysis is not complete: “The bivariate analysis was based on the analysis of variance, accompanied by pair-wise comparisons in post-hoc testing”. What specific tests and post-hoc test were performed? Pair-wise testing versus multiple comparison testing may not be appropriate.

Author Response

We would like to thank the reviewer for careful reading, identification of problems and suggestions provided. Also, thank you for picking up the main message, albeit buried it was; we have accepted and addressed all of your comments below.

The manuscript by Bućan and colleagues seeks to investigate whether there are any serum N-glycosylation patterns that could be used as a predictive biomarker for age-related macular degeneration (AMD). They compare the N-glycosylation patterns of a single molecule, IgG, between serum from 106 AMD patients, 45 pre-AMD patients, and 2684 control subjects. Yet, several conclusions of the paper do not appear to be fully supported by their results, as the underlying statistical analysis reveals very little, if any, real significant differences amongst conditions. My concerns are as follows:

  • While there appear to be some statistically significant differences amongst conditions in their initial analysis (that did not account for age or sex), these results do not seem particularly convincing in a biological context. For example, for many of these changes, the direction of change in early-onset drusen versus diagnosed AMD patients is completely different. If these changes were real, shouldn’t they change in the same direction? Nonetheless, many of their conclusions come from this analysis, whereas more focus should be put on the controlled analysis.

This is correct, and that is why we had consulted the senior statistician who suggested the use of a backwards regression model, capable to better detecting the associations that may be confounded by the high level of correlation between glycans. The results have slightly changed, and some glycans are now also significant in the adjusted models. The main conclusion holds that there are some changes in glycans that warrant further studies. We amplified this finding in the manuscript. Also, we would like to stress out that the reviewer had excellently picked up the idea of a non-linear disease progression; this is why we ask for an abandonment of the simple case-control approach but invoke the need for a more refined phenotype assessment, whatever the end-outcome might be. This approach seems to have shown some interesting results in this case, and maybe it holds similarly for other complex phenotypes.

  • In their second analysis, after accounting for age and sex, there are no significant differences between the controls and the early-onset drusen groups. Further, there are only two significant glycan changes in this analysis: IgG_GP18 for unilateral AMD and DG13 for bilateral AMD. Yet, the IgG_GP18 mark is no different in bilateral AMD, which casts doubt as to whether this difference is truly real.

R: sadly, the results for drusen have indeed become insignificant in the adjusted model, suggesting that we can’t really claim their significance. However, we do have to consider these results as at the very least interesting. Please do have in mind that the sample size is prohibitive for a finer analysis and that we indeed invoke the need to confirm this result in an unlinked study, which will include much better estimates in a prospective design.

  • The authors concede that “collapsing all three disease stages into a single outcome category (broad-sense AMD) did not yield a single significant result”. Yet, this appears to be buried, although this seems to be a key take-away. This should be bluntly stated, including in the abstract.

R: we thank the reviewer, and we do agree that this is an important take-away message – added to the Abstract

  • The authors make several conclusions to try to increase the significance of their results, which I do not find to be supported, such as:
    1. “The results suggested the anti-inflammatory changes in glycans in the early-stage drusen, breakdown of immune regulation in unilateral disease, and a substantial increase in the pro-inflammatory mechanisms in the bilateral disease occurrence”
    2. “The pattern of changes in this study revealed an interesting finding or lowered overall inflammatory signal”
    3. “Nevertheless, the results of this study demonstrate that serum or IgG glycans can be used as biomarkers for AMD progression”

R: All these instances were donw-tuned, to better reflect our findings 

  • The description of their statistical analysis is not complete: “The bivariate analysis was based on the analysis of variance, accompanied by pair-wise comparisons in post-hoc testing”. What specific tests and post-hoc test were performed? Pair-wise testing versus multiple comparison testing may not be appropriate.

R: corrected

Reviewer 3 Report

This is an interesting study about changes in IgG N-glycans observed in patients with age-related macular degeneration.  The work is sound, but I have the following remarks:

  1. The title sounds little awkward to me; would the Authors consider something more meaningful, e.g. "Changes in N-glycans"?
  2. Line 19: one cannot compare N-glycans to retinal fundus photographs. Also, the company name (Canon) belongs rather to the Materials and Methods, certainly not to the Abstract.
  3. Line 35: "sugary molecules" sounds awkward. Please re-write.
  4. The Authors should add here a paragraph about aetiology and pathogenesis of AMD.
  5. Line 80 and 82: "Therefore" appears two times.
  6. Line 91 and later: no explanation of statistcical terms, e.g. CI.
  7. Line 98: the Authors refer to the Supplementary Table, I was unable to find such table on the webpage. Maybe the Authors would consider to put the table in the main body?
  8. Line 162: "trans-organ similarity" is not the best term. Please re-write.

Author Response

We would like to thank the reviewer for careful reading, identification of problems and suggestions provided. We have accepted and addressed all of your comments below.

This is an interesting study about changes in IgG N-glycans observed in patients with age-related macular degeneration.  The work is sound, but I have the following remarks:

  1. The title sounds little awkward to me; would the Authors consider something more meaningful, e.g. "Changes in N-glycans"?

R: corrected, the title has been changed; notably, we removed the term changes, as it implied the existence of multiple measurements, and we performed a single one. The new title is now: The N-glycosylation patterns across the age-related macular degeneration spectrum.

  1. Line 19: one cannot compare N-glycans to retinal fundus photographs. Also, the company name (Canon) belongs rather to the Materials and Methods, certainly not to the Abstract.

R: corrected; we stated that we were exploring the association of glycans and retinal fundus photographs. Name of the camera was removed from the Abstract

  1. Line 35: "sugary molecules" sounds awkward. Please re-write.

R: removed the term „sugary“

  1. The Authors should add here a paragraph about aetiology and pathogenesis of AMD.

R: added, now starting at line 59

  1. Line 80 and 82: "Therefore" appears two times.

R: corrected

  1. Line 91 and later: no explanation of statistcical terms, e.g. CI.
  1. we added an explanation of the OR and CI abbreviations

  1. Line 98: the Authors refer to the Supplementary Table, I was unable to find such table on the webpage. Maybe the Authors would consider to put the table in the main body?

R: added, we apologize for the omission

  1. Line 162: "trans-organ similarity" is not the best term. Please re-write.

R: we removed that term

Round 2

Reviewer 2 Report

This reviewer appreciates the authors' response to my initial review. However, I still have a concern remaining regarding their overall conclusions about the study. The authors still claim in the abstract that “The results suggested the anti-inflammatory changes in glycans in the early-stage drusen, exhibited breakdown of immune regulation in unilateral disease, and a substantial increase in the pro-inflammatory mechanisms in the bilateral disease occurrence”. I believe this is an over-interpretation of their data from a study that only investigates glycan levels, but does not assess the role of these glycan changes in the immune response in AMD. This line should not be so definitively stated in the abstract, but rather left to a hypothetical discussion in the discussion section.

Author Response

Dear reviewer, thank you very much for the comment. We tried to downplay this by saying suggest vs. exhibited, but we see now that your comment is justified. We have therefore further relaxed the sentence, which now states: 

We detected several interesting differences across the analysed groups, suggesting that N-glycans can be used as biomarker for AMD. 

We have also slightly downplayed this in the limitations section, to further stress out the principal and exploratory nature of this study.